# Dysbindin Domain-Containing 1 in Prostate Cancer: New Insights into Bioinformatic Validation of Molecular and Immunological Features

**DOI:** 10.3390/ijms241511930

**Published:** 2023-07-25

**Authors:** Van Thi Ngoc Tram, Hoang Dang Khoa Ta, Gangga Anuraga, Phan Vu Thuy Dung, Do Thi Minh Xuan, Sanskriti Dey, Chih-Yang Wang, Yen-Nien Liu

**Affiliations:** 1International Ph.D. Program in Medicine, College of Medicine, Taipei Medical University, Taipei 110, Taiwan; ngoctramlabo@gmail.com; 2Department of Medical Laboratory, University Medical Center Ho Chi Minh City, Ho Chi Minh City 700000, Vietnam; 3Graduate Institute of Cancer Biology and Drug Discovery, College of Medical Science and Technology, Taipei Medical University, Taipei 110, Taiwan; d621109004@tmu.edu.tw (H.D.K.T.); g.anuraga@unipasby.ac.id (G.A.); phanvuthuydung@gmail.com (P.V.T.D.); m654110001@tmu.edu.tw (D.T.M.X.); m654111005@tmu.edu.tw (S.D.); 4Ph.D. Program for Cancer Molecular Biology and Drug Discovery, College of Medical Science and Technology, Taipei Medical University and Academia Sinica, Taipei 110, Taiwan; 5Department of Statistics, Faculty of Science and Technology, Universitas PGRI Adi Buana, Surabaya 60234, Indonesia; 6TMU Research Center of Cancer Translational Medicine, Taipei Medical University, Taipei 110, Taiwan

**Keywords:** prostate cancer, bioinformatic investigation, dysbindin protein family genes, *DBNDD1* gene expression, cell cycle, E2F, GSK-3β, microtubule cytoskeleton, immunology, tumor-infiltrating immune cell

## Abstract

Prostate cancer (PCa) is one of the most prevalent cancers in men, yet its pathogenic pathways remain poorly understood. Transcriptomics and high-throughput sequencing can help uncover cancer diagnostic targets and understand biological circuits. Using prostate adenocarcinoma (PRAD) datasets of various web-based applications (GEPIA, UALCAN, cBioPortal, SR Plot, hTFtarget, Genome Browser, and MetaCore), we found that upregulated dysbindin domain-containing 1 (*DBNDD1*) expression in primary prostate tumors was strongly correlated with pathways involving the cell cycle, mitotic in KEGG, WIKI, and REACTOME database, and transcription factor-binding sites with the *DBNDD1* gene in prostate samples. *DBNDD1* gene expression was influenced by sample type, cancer stage, and promoter methylation levels of different cancers, such as PRAD, liver hepatocellular carcinoma (LIHC), and lung adenocarcinoma (LUAD). Regulation of glycogen synthase kinase (GSK)-3β in bipolar disorder and ATP/ITP/GTP/XTP/TTP/CTP/UTP metabolic pathways was closely correlated with the *DBNDD1* gene and its co-expressed genes in PCa. *DBNDD1* gene expression was positively associated with immune infiltration of B cells, Myeloid-derived suppressor cell (MDSC), M2 macrophages, andneutrophil, whereas negatively correlated with CD8^+^ T cells, T follicular helper cells, M1 macrophages, and NK cells in PCa. These findings suggest that DBNDD1 may serve as a viable prognostic marker not only for early-stage PCa but also for immunotherapies.

## 1. Introduction

According to the American Cancer Society, prostate, lung, and colorectal cancers were estimated for nearly half of all male incidence cases in 2023, with PCa accounting for 29% of diagnoses [1]. PCa is one of the most common cancer types in males, but the pathogenic mechanisms of this disease remain poorly known [2]. The heterogeneity of PCa is comprised of variations in epidemiology and genetics. Survival rates for PCa vary by race due to the complex interaction of genetic, environmental, and social factors [3]. It causes differences in the epidemiology of the disease across countries [4]. Existing diagnostic techniques for PCa are invasive and lack the specificity to distinguish between aggressive and nonaggressive forms of the disease, frequently resulting in unnecessary overtreatment [3]. In the present work, we sought to identify a relevant genetic biomarker of PCa to develop and validate the potential risks of this disease by incorporating the prostate-specific antigen (PSA) and other existing techniques. Bioinformatic analyses utilizing transcriptomics and high-throughput sequencing can aid in the identification of specific cancer diagnostic biomarkers and gain a deeper understanding of molecular pathways [5]. Similarly, machine learning techniques can aid in identifying transcripts that correlate with the progression of PCa. Several potential biomarkers were examined to predict PCa progression, particularly the disease’s stages [6]. Now that computational and next-generation sequencing technologies have advanced, it is possible to investigate patients’ genomic profiles in conjunction with their Gleason scores more precisely and effectively [7].

The dysbindin protein has multiple physiological functions, especially in the nervous system [8]. In humans, mice, and rats, at least three paralogs of the dysbindin protein family were identified with the designations dysbindin-1, dysbindin-2, and dysbindin-3 [9]. The dystrobrevin-binding protein 1 gene (*DTNBP1*), located at position 22.3 on the short arm (p) of chromosome 6, is responsible for encoding the protein dysbindin-1. In the central nervous system (CNS), the dysbindin-1 protein preserves the structure and physical stability of neuronal synaptic membranes [10,11]. Three paralogs are expressed in the brain, but dysbindin-1 has garnered the most attention due to polymorphisms in its encoding gene, *DTNBP1*, and controversies around its relationship with schizophrenia, and cognitive deficits [12,13,14]. Dysbindin-2 is expressed in neurons and glia of white matter tracts in mammals, most likely oligodendrocytes. Dysbindin-2 is encoded by the *DBNDD2* gene located at 20q13.12 on the human chromosome. Dysbindin-3 is encoded by the *DBNDD1* gene at human chromosome 16q24.3 [9]. 

Several malignancies are associated with dysbindin. Dysbindin may promote tumor growth in pancreatic cancer by activating the phosphatidylinositol 3-kinase (PI3K)/AKT signaling pathway, and dysbindin overexpression is correlated with a poor prognosis for pancreatic ductal adenocarcinoma [15]. A higher DTNBP1 level is associated with shorter overall survival (OS), and a lower DTNBP1 level reduces cell proliferation and induces apoptosis in hepatocellular carcinoma cell lines [16]. Dysbindin in epithelial ovarian cancer increases tumor cell invasion and metastasis by activating extracellular signal-regulated kinase (ERK) phosphorylation and triggering the epithelial-mesenchymal transition [17]. To our knowledge, there is limited evidence investigating the role of dysbindin in advancing PCa. Interestingly, we found that only *DBNDD1* gene expression in the dysbindin protein family was substantially elevated in prostate tumor samples, whereas *DTNBP1* and *DBNDD2* gene expressions were not. Using the PRAD datasets that are parts of the Gene Expression Profiling Interactive Analysis (GEPIA) and web-based applications (UALCAN, Enrichr, Xenabrowser, SR Plot, hTFtarget, Genome Browser, and MetaCore) that provide rapid and customizable functionalities based on The Cancer Genome Atlas (TCGA), The Common Fund’s Genotype-Tissue Expression (GTEx), and chromatin immunoprecipitation (ChIP)-sequencing (Seq) data [18], we were able to determine *DBNDD1* gene expression levels and then investigate biological processes associated with the *DBNDD1* gene in PCa development.

Androgen-deprivation therapy (ADT) has become one of the main treatments for advanced PCa cases or those who can not be cured by surgery. However, ADT also induces many adverse side effects, including metabolic dysfunction, insulin resistance, an increase in the fat mass, sexual dysfunction, a deterioration in the quality of life, and others [19,20]. Virtually all patients treated with ADT exhibit a transitory progression from hormone-sensitive PCa to castration-resistant PCa (CRPC); approximately 10% of CRPC cases progress to neuroendocrine PCa (NEPC) [21]. To support and counteract the unfavorable effects of ADT, it is necessary to develop secondary therapeutic options for PCa patients. In practice, most individuals with PCa are resistant to immunotherapy, and this was particularly true in trials using immune checkpoint inhibitors [22]. Recent research suggests that interactions between tumor cells and the host’s immune system affect the onset, progression, and development of CRPC. Due to the distinct functions of subpopulations of immune cells, it is crucial to comprehend how prostate cells interact with the functional characteristics of immune cells, such as B cells, cluster of differentiation 4-positive (CD4^+^) T cells, CD8^+^ T cells, macrophages, myeloid-derived suppressor cells (MDSCs), and others [23]. In our study, we attempted to determine links between the *DBNDD1* gene and networks regulated by transcription levels and immune infiltration in PCa to assist in identifying a new potential immunotherapeutic target in the prostate tumor microenvironment (TME). To explain transcriptional levels, molecular structures, functional enrichment analyses, and immune infiltration, a comprehensive investigation of genes encoding the dysbindin protein family in specific cancer candidates, particularly in PCa, is required (Figure 1).

## 2. Results

### 2.1. Expressions of Genes Encoding Dysbindin Protein Family Members in a Pan-Cancer Analysis

To determine how genes encoding members of the dysbindin protein family are expressed in various cancers, we searched the TNM plot and TCGA datasets. We performed a pan-cancer analysis with representative genes encoding dysbindin-1, dysbindin-2, and dysbindin-3, including the *DTNBP1, DBNDD2,* and *DBNDD1* genes, respectively. *DTNBP1* gene expression was significantly increased in a variety of cancers, including acute myeloid leukemia (AML), breast cancer, colon cancer, liver cancer, lung cancer, ovary cancer, pancreatic cancer, rectal cancer, renal cancer, skin cancer, thyroid cancer, and uterine cancer (Figure 2A). However, expression of the *DTNBP1* gene did not significantly differ between normal and tumor tissues in PCa, testicular cancer, and other cancers. While expression of *DBNDD2* messenger (m) RNA was lower in all cancer types (Figure 2B), expression of the *DBNDD1* gene was significantly higher in numerous tumor types, including PCa, lung cancer, and liver cancer, among others (Figure 2C). In addition, results of the analysis of *DBNDD1* gene expression was confirmed by another dataset in the GEPIA web-based tool (Figure 2D). In our work, we focused primarily on analyzing *DBNDD1* expression in the most prevalent male cancers; therefore, PCa, liver cancer, and lung cancer were chosen for further examination, and they also showed similar ratios of fold changes in *DBNDD1* expression between normal and tumor tissues. 

To understand clinical characteristics associated with *DBNDD1* expression, we examined whether sample type, cancer stages, and promoter methylation levels influenced *DBNDD1* expression in the three abovementioned cancers. *DBNDD1* mRNA levels were significantly higher in three primary PRAD, LIHC, and LUAD tumors relative to their respective normal groups (Figure 2E). During the past 50 years, Gleason scores (ranging from 6 to 10) have been used to forecast and guide treatment for PCa patients, with higher scores indicating a more significant risk of disease than an intermediate risk of disease [24]. *DBNDD1* expression profiles were significantly elevated in samples with Gleason scores of 6 to 10 compared with normal samples (Figure 2F, left). Likewise, its expression was considerably upregulated in aberrant stages 1 to 4 in LIHC and LUAD (Figure 2F, middle and right). DNA methylation is an epigenetic process that regulates gene expressions by altering interactions of chromatin proteins and transcription factors with DNA. Changes in promoter methylation may be a defining characteristic of cancer related to the silencing of tumor-suppressor genes and activation of oncogenes [25]. That is consistent with our findings that levels of abnormal promoter methylation of *DBNDD1* were dramatically decreased in PRAD, LIHC, and LUAD (Figure 2G). Our preliminary results revealed that expression of the *DBNDD1* gene was significantly affected by sample type, cancer stages, and promoter methylation levels of different cancers such as PRAD, LIHC, and LUAD.

**Figure 2 ijms-24-11930-f002:**
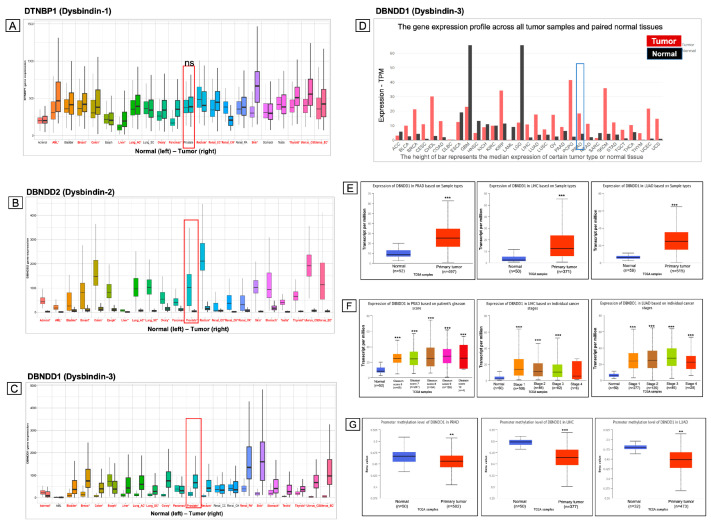
Dysbindin protein family gene expression is analyzed in various cancer types. A pan-cancer analysis was performed on the TNM plot and TCGA datasets. (**A**) High *DTNBP1* expression was observed in many different types of cancers. (**B**) The *DBNDD2* gene was downregulated in all cancer types. (**C**,**D**) High *DBNDD1* expression was found in many types of cancers, as confirmed by the datasets in TNM plot and GEPIA web tools. TNM plot, significant differences determined by the Mann-Whitney U test marked in red. GEPIA, Log Scale is log_2_(TPM + 1) transformed expression data. (**E**) *DBNDD1* expression was correlated with sample types in prostate adenocarcinoma (PRAD), liver hepatocellular carcinoma (*LIHC*), and lung adenocarcinoma (LUAD) (UALCAN). (**F**) DBNDD1 expression was correlated with Gleason scores and cancer stages in PRAD, LIHC, and LUAD (UALCAN). (**G**) DBNDD1 expression was associated with promoter methylation levels in PRAD, LIHC, and LUAD (UALCAN). Student’s *t*-test was used for statistical tests between groups, (*) indicates a significant difference with *p* < 0.05, (**) indicates a significant difference with *p* < 0.01. (***) indicates a significant difference with *p* < 0.001, (ns) indicates non-significant.

### 2.2. Survival Analysis of DBNDD1 Gene Expression

After identifying the *DBNDD1* transcriptomic level, we investigated relationships of *DBNDD1* gene expression, with corresponding tumor levels of patients and disease prognostication in PRAD, LIHC, and LUAD. *DBNDD1* gene expression was significantly elevated in PRAD, LIHC, and LUAD tumor tissues compared with normal tissues based on datasets of the TNM plot and GEPIA-based tool (Figure 3A). However, *DBNDD1* gene expression was only statistically significant in PRAD (Figure 3A, left) but not in LIHC or LUAD metastatic tissues (Figure 3A, middle and right). Next, we used the GEPIA web application to examine survival plots from a PCa database. The curve analysis and log-rank test revealed that greater levels of *DBNDD1* mRNA expression were only substantially linked with poor OS in PRAD (log-rank *p* = 0.049) (Figure 3B, left). There were no statistically significant associations between LIHC (log-rank *p* = 0.32) or LUAD (log-rank *p* = 0.63) (Figure 3B, middle and right). Higher levels of *DBNDD1* mRNA expression were correlated with poor disease-free survival (DFS) in PRAD (log-rank *p* = 0.099), but the correlation was not statistically significant (Figure 3C, left). Also, *DBNDD1* expression was not associated with the DFS rate in LIHC or LUAD (Figure 3C, middle and right). Regarding tumor stages, metastasis, and OS rates, we found the effect of *DBNDD1* gene expression to be more significant in PCa than in liver and lung cancers. Furthermore, we focused on the following investigation of *DBNDD1* gene expression in PCa.

### 2.3. Differential Expression Analysis Identifies Upregulation of the DBNDD1 Gene in PCa

To investigate DBNDD1 alterations involved in PCa, we next analyzed gene expression signatures of prostate tumor tissue samples versus solid normal tissue samples using GSE datasets. To validate our findings, we utilized alternative cancer databases on patient cohorts, such as GSE3325 (genomic and proteomic integration of primary and metastatic cancer samples in PCa patients relative to benign prostate individuals) and GSE55945 (identification of candidate biomarkers and immunotherapy targets based on genome-wide gene expression in benign and malignant prostate tissues). We performed a differential expression analysis of quantitative genomic data using the log2FC ± 0.5-fold change thresholds in PCa over the control (*p* < 0.05). Then we identified 2828 differentially expressed genes (DEGs) (1009 downregulated genes and 1819 upregulated genes), and the *DBNDD1* gene was significantly upregulated in primary PCa tissues (Figure 4A). In addition, *DBNDD1* gene expressions were substantially increased in both primary PCa patients (*p =* 0.014) and metastatic PCa patients (*p* = 0.026) compared with the control cohort (Figure 4B). These alterations were consistent with the results of GSE55945 (Figure 4C,D). Next, we validated *DBNDD1* expression in GSE220942 (the role of GALNT7 upregulation in PCa cells) and GSE217260 (DEGs of benign prostatic hyperplasia and high-risk prostatic carcinoma tissues). Consequently, DBNDD1 expression was significantly elevated in GALNT7-overexpressing DU145 PCa cells and high-risk prostatic tissues relative to control samples (Figure 4E,F). These results demonstrated that DBNDD1 expression was considerably increased in the PCa progression compared with benign conditions. 

In the HPA database, we observed protein expressions of DBNDD1 in PCa patients using IHC staining, along with clinicopathological data such as patient ID, age, and gender in normal (Figure 4G) and tumor (Figure 4H) tissue samples. Overexpression of DBNDD1 protein levels was detected in prostate tumor tissue samples from the HPA database. These outcomes corresponded with mRNA *DBNDD1* expression profiles presented in previous results. A pie graph depicts the IHC staining intensity of the DBNDD1 protein in PCa samples (Figure 4I). In addition, expression of the *DBNDD1* gene was analyzed in different PCa cell lines using the CCLE database. Results revealed that the MDAPCA2B, DU145, LNCAP, and NCIH660 cell lines had significant levels of *DBNDD1* expression (Figure 4J). We also noted each cell line’s androgen receptor (*AR*) expression. We investigated the interaction between the *DBNDD1* and *AR* genes in PCa using RNA-Seq data. We found that *DBNDD1* expression was positively correlated with the *AR* in normal prostate tissues but not significant in tumor prostate tissues (Appendix A).

Next, we examined the mRNA levels of *DBNDD1* in various PCa cell lines using the RT-qPCR method. Interestingly, results demonstrated that *DBNDD1* mRNA expression was significantly higher in LNCaP and C4-2 cells compared with PZ-HPV7 normal prostate cells than in LASCPC-01 and PC3 cells (Figure 4K), and this difference was confirmed in LNCaP and PC3 cells by repeating the experiment (Figure 4L). It suggests that *DBNDD1* is more abundant in AR-positive PCa cells than in AR-negative or AR-independent PCa cells, and *DBNDD1* may be affected by AR regulation.

**Figure 4 ijms-24-11930-f004:**
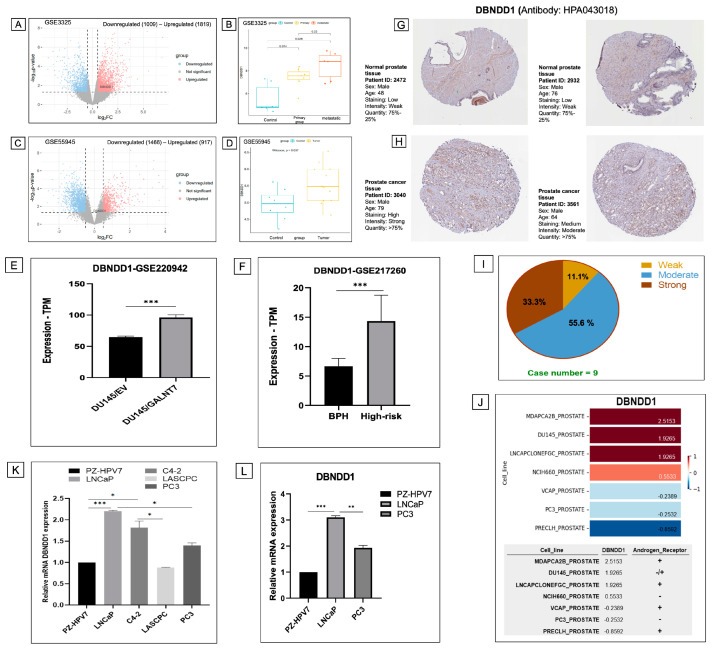
DBNDD1 expression was upregulated in prostate cancer progression. (**A**,**C**) Volcano plots depicting distributions of the *DBNDD1* gene and other genes in primary tumor tissues vs. solid normal tissues in GSE3325 and GSE55945 datasets. Upregulated genes with log2FC > 0.5 and *p* < 0.05 are in red, and downregulated genes with log2FC < −0.5 and *p* < 0.05 are in blue. (**B**,**D**) Box plots depicting *DBNDD1* gene expression in control, primary tumor, and metastatic tissues of individuals in GSE3325 and GSE55945 datasets. Wilcoxon test was used for statistical tests between groups, (*) indicates a significant difference with *p* < 0.05, (**) indicates a significant difference with *p* < 0.01, (***) indicates a significant difference with *p* < 0.001, (ns) indicates non-significant. (**E**) *DBNDD1* expression was increased in GALNT7-overexpressing DU145 prostate cancer cells compared with control cells (GSE220942). (**F**) *DBNDD1* expression was increased in high-risk prostatic tissues compared with benign prostatic hyperplasia (BPH) tissues (GSE217260). Student’s *t*-test was used for statistical tests between groups with statistical significance at *p* < 0.05 (*); *p* < 0.01 (**); *p* < 0.001 (***). (**G**,**H**) IHC-stained pictures displayed the intensities of antibodies in adjacent normal tissues and prostate cancer tissues. (**I**) A pie chart indicates the IHC staining intensity of the DBNDD1 protein in 9 cases. (**J**) The data table displays *DBNDD1* gene mRNA levels in various prostate cancer cell lines (CCLE) and androgen receptor expression in each cell line. (**K**,**L**) *DBNDD1* mRNA expression was shown in LNCaP, C4-2, LASCPC-01, PC3 cells compared with PZ-HPV7 normal prostate epithelial cells, measured by an RT-qPCR analysis. Student’s *t*-test was used for statistical tests between groups with statistical significance at *p* < 0.05 (*); *p* < 0.01 (**); *p* < 0.001 (***).

### 2.4. Upregulated DBNDD1 Expression Was Strongly Correlated with Cell Cycle Process

Next, we performed an enrichment analysis with the Enrichr web-based tool to find gene ontology (GO) keywords and potential pathways associated with the *DBNDD1* gene and other upregulated genes in PCa. Regulation of the cell cycle process and cell division, mitotic sister chromatid segregation, mitotic spindle organization, mitotic nuclear division, and positive regulation of ubiquitin-protein transferase activity were the most important biological activities (Figure 5A). The most significant functionalities associated with GO molecular functions were protein and cadherin bindings involved in cell-cell adhesion (Figure 5B). Moreover, the cellular component analysis revealed specific localization in chromosome structures: condensed chromosomes, centromeric regions, condensed chromosome kinetochores, microtubules, the microtubule cytoskeleton, nuclear ubiquitin ligase complexes, anaphase-promoting complexes, and fibrillar centers (Figure 5C). Furthermore, upregulated *DBNDD1* expression in primary prostate tumors was strongly correlated with the “cell cycle” in KEGG pathways (Figure 5D), “Gastric cancer Network 1 WP2361”, “ApoE and miR-146 in inflammation and atherosclerosis WP 3926”, and “cell cycle WP179” in WIKI pathways (Appendix A), and issues involved in the cell cycle and mitotic *Homo sapiens* in REACTOME pathways (Appendix A).

We also investigated the mutation characteristics of *DBNDD1* in PRAD from a TCGA cohort using the cBioPortal tool. We determined that the highest genetic alteration frequency of *DBNDD1* in PRAD was roughly 5% compared with other types of carcinoma (Figure 5E). In addition, the predominant type of genetic alterations (of >4.5% frequency) in PRAD cases was copy number deletions of *DBNDD1*. In contrast, mutations accounted for fewer than 0.5% of cases. We visualized multiple genomic alteration events of the *DBNDD1*, *DBNDD2*, and *DTNBP1* genes across a set of prostate tumor samples using OncoPrint using a query for alterations (Figure 5F). For *DBNDD1*, most of the alterations were deep deletions. A few modifications in *DBNDD2* were amplifications, and few events related to *DTNBP1* occurred. Most analyzed cancers exhibited high numbers of *DBNDD1* genetic alterations of amplifications and copy number deletions, except lung squamous cell carcinoma which had complete mutations. 

**Figure 5 ijms-24-11930-f005:**
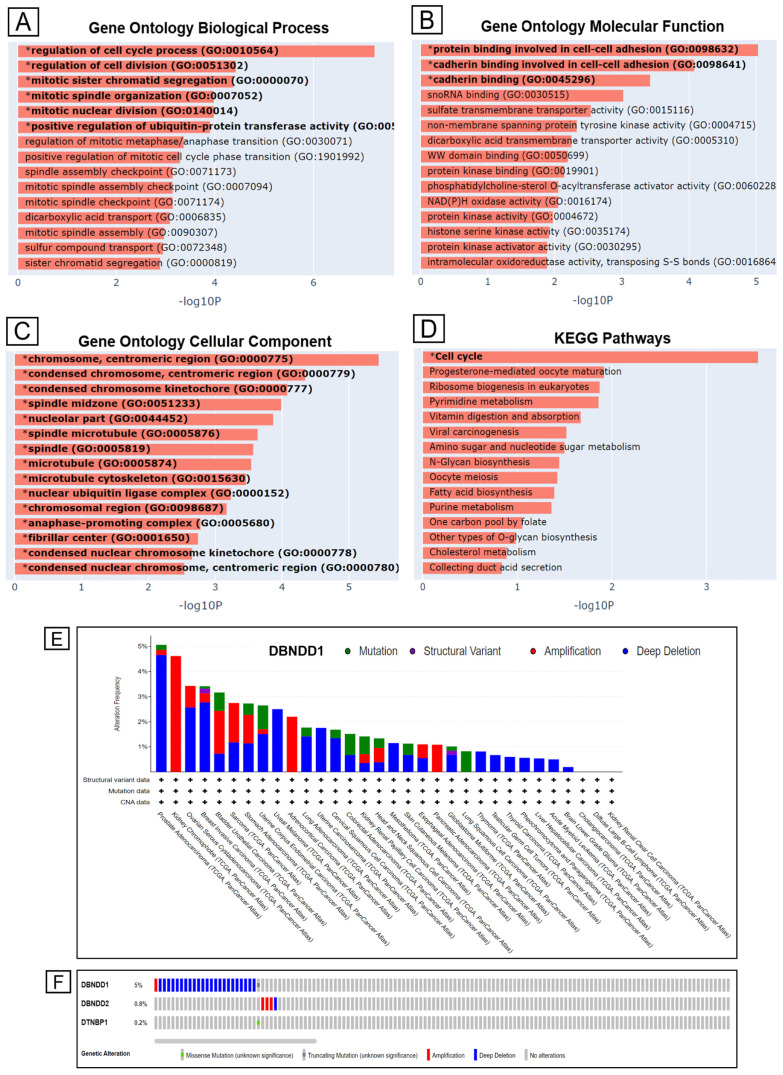
Gene ontology (GO) enrichment results for primary tumor tissues vs. solid normal tissues in TCGA prostate cancer differential gene expression analysis. Genomic alterations of *DBNDD1* in prostate adenocarcinoma (PRAD) from TCGA cohorts using the cBioPortal web tool. (**A**–**C**) GO keywords with significant gene *p* values, biological process (**A**), molecular functions (**B**), and cellular components (**C**) for the *DBNDD1* gene and upregulated genes. (**D**) Result of an enrichment analysis including KEGG pathways. Important phrases are emphasized in boldface font. Significantly enriched GO and KEGG analyses (*) are shown with Benjamini Hochberg false detection rate (FDR)-corrected *p* values. (**E**) Alteration frequencies with mutation type of *DBNDD1* in various tumor samples from TCGA cohorts using cBioPortal. (**F**) The tab shows genomic alterations in the *DBNDD1, DBNDD2,* and *DTNBP1* genes across a sample set. Each row represents a gene; each column represents a tumor sample. Gene amplifications are represented by red bars, deep deletions by blue bars, and missense mutations by green squares.

### 2.5. DBNDD1 Gene Expression and Transcription Factor-Binding Sites

To better understand transcription factor-target regulation of gene expression, we used the Enrichr web-based tool to search for upstream transcription factors that regulate the top 273 genes co-expressed with *DBNDD1*. In the “TRRUST Transcription Factors 2019” database of transcription regulatory interactions; the AR displayed a lower adjusted *p* value (0.038) (Figure 6A). Next, we looked into how the AR transcription factor and *DBNDD1* target regulation in humans using ChIP-Seq data and transcription factor-binding sites scanned using the web tool of human transcription factor target (hTFtarget) [26]. According to datasets of ChIP-Seq samples in which human transcription factors were collected, we found the AR transcription factor in kidney and prostate tissues with peaks of the strongest signals (Chr16, 90020947, 90021454, 15.3, −132, pr, dataset-67, and Chr16, 90020000, 90020431, 52.6, −303, pr, dataset-3875). AR-DBNDD1 regulation was shown in kidney tissues with visualized peak information in the genome browser (Figure 6B). However, visualization of prostate tissues did not support the database analysis of the hTFtarget web tool; therefore, we could not collect the visualized genome browser.

Next, we investigated how *DBNDD1* mRNA levels interact with AR and AR-responsive genes (KLK3, NKX3-1, and TMPRSS2) in different PCa cells. As a result, *DBNDD1* mRNA expression was upregulated with *KLK3*, *NKX3*-*1*, and *TMPRSS2* in LNCaP cells relative to PZ-HPV7 control cells. Conversely, these genes were substantially downregulated in PC3, an AR-independent PCa cell line, relative to LNCaP cells (Figure 6C, left). Additionally, we found that AR inhibition (treatment with MDV3100) decreased the mRNA levels of *DBNDD1*, *KLK3*, *NKX3-1*, and *TMPRSS2* in LNCaP cells. It indicated that *DBNDD1* expression is affected by the regulation of AR in PCa (Figure 6C, right). Using Genome Browser (Genomics Institute, UCSC, Santa Cruz, CA, USA) results revealed 161 transcription factors from ENCODE with Factorbook Motifs, which bound to the promoter region of the *DBNDD1* gene in prostate samples such as EBF1, PAX5, BCL11A, EBF1, MYC, FOXA1, FOXA2, CEBPB, and others (Figure 6D).

**Figure 6 ijms-24-11930-f006:**
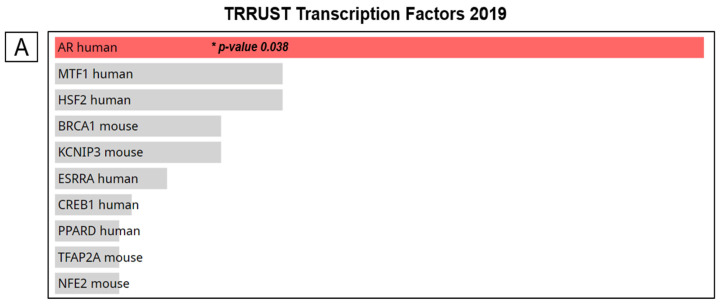
Regulation of transcription factor-binding sites and DBNDD1 gene expression. (**A**) Bar graph of “Transcription Factors 2019” depicting the top ten enriched transcription factors of the chosen genes. The bars are ordered according to their p values, with red bars indicating statistical significance. (**B**) Androgen receptor (AR)-DBNDD1 regulation was demonstrated in kidney tissues using the genome browser’s visible peak information. (**C**) Relative DBNDD1, AR, NKX3-1, KLK3, and TMPRSS2 mRNA levels were expressed in LNCaP, PC3 cells compared with PZ-HPV7 normal epithelial prostate cells, in MDV3100-treated LNCaP compared with DMSO-treated LNCaP cells, measured by an RT-qPCR analysis. Student’s *t*-test was used for statistical tests between groups, (*) indicates a significant difference with *p* < 0.05, (**) indicates a significant difference with *p* < 0.01, (***) indicates a significant difference with *p* < 0.001, (ns) indicates non-significant. (**D**) DNA-binding motifs were obtained from the ENCODE Factorbook storage facility, a set of all ENCODE transcription factor ChIP-Seq datasets. The extent of darkness correlates with the signal strength, and green highlights represent the highest-scoring motif sites.

### 2.6. High Expression of DBNDD1 Is Associated with E2F Transcription Factor Targets and Mitotic Spindle Checkpoint Signaling in PCa

It was exciting to understand more about the molecular mechanisms underlying the co-expression of *DBNDD1* with gene sets. Enrichment of the Molecular Signature Database (MSigDB_Hallmark gene sets) in PCa samples with high *DBNDD1* expression was determined using a GSEA. E2F is known to regulate the cell cycle and had the highest normalized enrichment score (NES) of 2.147 (Figure 7A). E2F transcription factors (E2Fs) consist of eight members: E2F1 to E2F8. Recent research revealed the E2F-targeted pathway is associated with high SLC35A2 expression in breast cancer [27]. The E2F pathway score strongly correlated with expressions of cyclin-dependent kinase pathway-related genes and immune checkpoint molecules, and had a favorable link with the responsiveness of cyclin-dependent kinase inhibitors [28]. Through modulating CD147, E2F1 increased the invasion and migration of PCa cells, and significantly, E2F1 overexpression forecasted a poor prognosis for human PCa [29]. E2F3 directly targets interleukin (IL)-6 signaling and is associated with prostate carcinogenesis [30]. Disruption of the E2F5/p38/SMAD3 network was reported to enhance the protumorigenic transition of transforming growth factor (TGF) signaling in PCa [31]. E2F7, controlled by miR-30c, suppresses apoptosis and increases cell cycle progression in PCa cells [32]. However, the effects of E2F2, E2F4, E2F6, and E2F8 on prostate etiology and cancer growth are poorly understood. Using RNA-Seq data, we determined interactions between DBNDD1 expression and E2Fs in PCa. Consequently, E2F1, E2F4, and E2F5 had negative correlations with DBNDD1 expression in normal prostate tissues but positive correlations in prostate carcinoma tissues (Appendix A). E2F2 was positively correlated with DBNDD1 expression in both normal and tumorous prostate tissues, whereas E2F6 was negatively correlated with DBNDD1 expression in both tissue types. E2F3 was negatively correlated with DBNDD1 expression in normal prostate tissues but showed no significance in tumor tissues. E2F7 expression was the opposite of E2F3. E2F8 lacked statistical importance in both tissue types (Appendix A). 

Moreover, DBNDD1 was observed to be highly expressed in a group of cancer-involved factors, including the G_2_M checkpoint (Appendix A), mitotic sister chromatid segregation (Figure 7B), and mitotic spindle assembly checkpoint signaling (Figure 7C). The GSEA revealed hallmark analysis signaling pathways significantly associated with the co-expression of *DBNDD1* with gene sets including glycolysis (Appendix A), PI3K-AKT-mammalian target of rapamycin (mTOR) signaling (Appendix A), and DNA repair (Appendix A). Furthermore, using ChIP-seq data from studies of Hes6-AR-E2F1 interaction (GSE49832) and RB loss-E2F1 interaction (GSE94958) in LNCaP cells, we determined that AR ligand treatment (bicalutamide) of LNCaP cells may result in E2F1 binding to the transcriptional start site of the *DBNDD1* gene. The absence of RB may enhance overall E2F1 binding capacity and E2F1 linkage with the *DBNDD1* transcriptional start site (Figure 7D). RB may have some functions involving DBNDD1 expression. To better understand the relationship between E2F1, DBNDD1, and AR, we examined their mRNA levels in E2F1-overexpressing LNCaP and C4-2 cells compared with empty vector (EV) cells. The E2F1 overexpression significantly increased the mRNA levels of AR-responsive genes and slightly increased the mRNA levels of *DBNDD1* in LNCaP cells (Figure 7E). Likewise, the E2F1 overexpression significantly induced mRNA *DBNDD1* expression in C4-2 cells (Figure 7F). These results indicate that activated E2F1 may bind directly to the *DBNDD1* gene in PCa cells.

**Figure 7 ijms-24-11930-f007:**
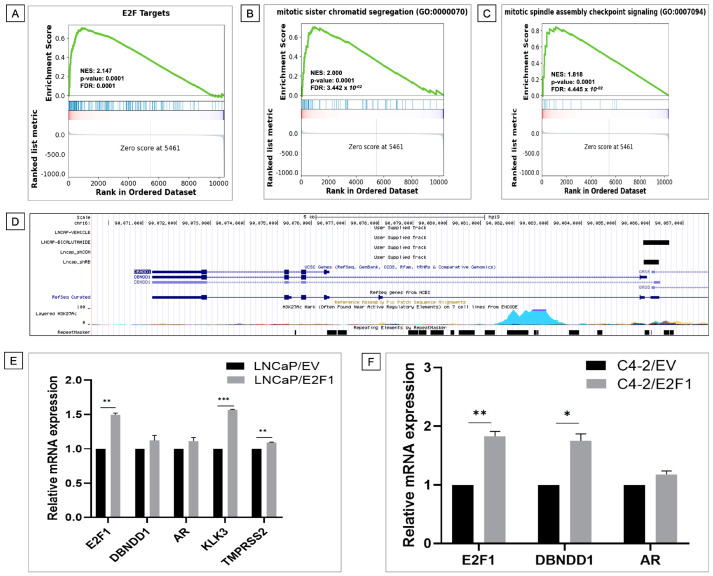
Results of a gene set enrichment analysis (GSEA) in prostate cancer patients in TCGA cohort with high *DBNDD1* expression. (**A**–**C**) Patients were separated into two groups based on their *DBNDD1* mRNA expression levels in TCGA PRAD dataset; afterward, a gene ranking list was produced and entered into the GSEA. As indicated by the GSEA database, statistical significance was evaluated using a false detection rate (FDR) value of 0.25, a normalized enrichment score (NES) of more than 1.3, and a nominal *p*-value of 0.05. Enrichment at the top of the list is indicated by a positive NES value, which suggests the enrichment pathway. (**D**) A ChIP-sequencing analysis of the detected DNA-binding sites for the E2F1 of the *DBNDD1* gene in cells in LNCaP-shCON and LNCaP-shRB cells (GSE94958-GSM2492420-GSM2492421), and LNCaP cells with and without bicalutamide (GSE49832-GSM1207897-GSM1207901). ChIP-sequencing data were obtained from the Gene Expression Omnibus (GEO) and analyzed by Genome Browser. (**E**,**F**) Relative *E2F1*, *DBNDD1*, *AR*, *KLK3*, and *TMPRSS2* mRNA levels were shown in LNCaP and C4-2 cells transfected with an empty vector (EV) and *E2F1*-overexpressing vector, measured by an RT-qPCR analysis. Student’s *t*-test was used for statistical tests between groups, (*) indicates a significant difference with *p* < 0.05, (**) indicates a significant difference with *p* < 0.01, (***) indicates a significant difference with *p* < 0.001, (ns) indicates non-significant.

### 2.7. DBNDD1 Exhibits an Essential Role in the Regulation of Glycogen Synthase Kinase (GSK)-3β in Bipolar Disorder

We used MetaCore software (https://portal.genego.com accessed on 24 February 2023) to investigate downstream networks associated with the previously reported co-expression patterns of the *DBNDD1* gene. We extracted and selected the top 273 genes (Spearman’s value of ≥0.4) in the *DBNDD1* co-expression profiles from TCGA dataset (Appendix A). Consequently, GeneGo MetaCore’s annotations of biological processes (Appendix A) revealed that *DBNDD1* was significantly associated with a number of metabolic pathways, including the highest ranking of “regulation of GSK3-beta in bipolar disorder” (Figure 8), “ATP/ITP metabolism” (Appendix A), and “GTP-XTP metabolism” (Appendix A). GSK-3 consists of two isoforms, GSK-3α and GSK-3β, which are part of the family of serine/threonine kinases found in all eukaryotic organisms, and this is the enzyme that catalyzes the final step in glycogen synthesis [33]. As depicted in the schematic, axin interacts with GSK-3 and is crucial for GSK-3-dependent regulation of the canonical WNT signaling pathway. The NGF, FGF2, NT-3, BDNF, IL-1-β, and dopamine factors also indirectly impact the action of GSK-3β in bipolar disorder.

### 2.8. Levels of Immune Infiltration Were Associated with DBNDD1 Expression in PCa

The TIMER database was used to investigate associations between immune infiltration and *DBNDD1* expression in PRAD in the immunological microenvironment. We found that *DBNDD1* gene expression was positively correlated with immune infiltration of B cells(*r* = 0.272, *p* = 1.79 × 10^−8^), purity (*r* = 0.204, *p* = 2.79 × 10^−5^), CD4^+^ effector memory T cells (*r* = 0.142, *p* = 3.77 × 10^−3^), CD4^+^ type 1 helper T cells (*r* = 0.198, *p* = 4.98 × 10^−5^), M2 macrophages (*r* = 0.118, *p* = 1.65 × 10^−2^), and myeloid-derived suppressor cells (*r* = 0.197, *p* = 5.28 × 10^−5^), mast cells (*r* = 0.212, *p* = 1.25 × 10^−5^), and neutrophils (*r* = 0.108, *p* = 2.08 × 10^−2^) in PRAD (Figure 9A). The function of B cells in cancer immunity can differ depending on the type of cancer [34]. B cells can enhance the growth of skin carcinomas [35] and induce immunosuppressive effects by activating inhibitory Fc receptors on myeloid cells [36]. However, enhancing B cell function can generate lymphotoxin, an inflammation-responsive IkB kinase α-activating cytokine, that boosts the survival of androgen-deprived PCa [37,38]. B and CD4^+^ T cells are crucial for developing immune-based treatments for all disease subtypes [39,40]. On the other hand, *DBNDD1* gene expression was negatively correlated with immune infiltration of CD8^+^ T cells (*r* = −0.204, *p* = 2.66 × 10^−5^), Treg cells (*r* = −0.171, *p* = 4.74 × 10^−4^), follicular helper T cells (*r* = −0.105, *p* = 3.19 × 10^−2^), natural killer (NK) T cells (*r* = −0.165, *p* = 7.31 × 10^−4^), monocytes (*r* = −0.286, *p* = 2.96 × 10^−9^), M1 macrophages (*r* = −0.196, *p* = 5.58 × 10^−5^), myeloid dendritic cells activated (*r* = −0.137, *p* = 5.25 × 10^−3^), natural killer (NK) cells (*r* = −0.17, *p* = 4.98 × 10^−4^), endothelial cells (*r* = −0.146, *p* = 2.8 × 10^−3^), cancer-associated fibroblasts (*r* = −0.301, *p* = 3.85 × 10^−10^), and hematopoietic stem cells (*r* = −0.272, *p* = 1.66 × 10^−8^) (Figure 9B). According to details of immune level algorithms, *DBNDD1* gene expression was positively and significantly associated with B cells by plasma XCELL (*r* = 0.27, *p* = 1.79 × 10^−8^), MPC COUNTER (*r* = 0.143, *p* = 3.58 × 10^−3^), QUANTISEQ (*r* = 0.127, *p* = 9.63 × 10^−3^), and Class-switched memory XCELL (*r* = 0.1, *p* = 0.04). Similarly, CD4^+^ effector memory T cells, M2 macrophages, myeloid-derived suppressor cells, and others, were also demonstrated by immune level algorithms (Appendix A).

## 3. Discussion

PCa is referred to as a heterogeneous illness since its formation is poorly understood at the molecular and genetic levels. Its pathophysiology is governed by molecular pathways strongly linked with the survival, metabolic, and metastatic characteristics of malignancies [41]. Using various databases, we can determine that *DBNDD1* gene expression was elevated in numerous types of cancer, and it was associated with prognosis and immune infiltration in PCa patients. Methylation levels of the promoter of DBNDD1 were considerably reduced in PCa (Figure 2G). Gene promoter methylation is a well-known epigenetic process that occurs during the early phase of tumor formation; hence, it is regarded as a possible biomarker for cancer diagnosis [42]. However, the precise mechanism and function of these DNA methylation alterations in cancer remain unknown [25]. To our knowledge, previous investigations have yet to identify the roles of dysbindin protein family genes in PCa. This is the first study to employ bioinformatics and data mining to examine transcription levels and biological functions of the *DBNDD1* gene in PCa. DBNDD1 expression was elevated in PCa cell lines, PRAD patients, had irregular promoter methylation, and was linked with PRAD patient survival times, primary tumors, and various stages. These findings strongly imply that DBNDD1 may be a viable prognostic marker for PRAD in its early stages.

Surgery, radiation, and emerging immunotherapies are the most prevalent treatments for early stage PCa. ADT is usually reserved for individuals who cannot undergo curative surgery or have high-risk locally or systemically advanced cancer [43]. Metastasis, however, might become the leading cause of mortality for men with PCa when aggressive cancer cells spread to other organs. Patients can acquire fatal CRPC after prolonged ADT [2]. Hence, a prompt and precise diagnosis is essential for selecting the most effective treatments. The introduction of microarray and sequencing technologies has benefited oncological research through gene expression analyses and comprehensive bioinformatics analyses using various datasets. In our GSEA findings from the TCGA PRAD dataset, *DBNDD1* gene expression was positively correlated with the mitotic spindle checkpoint of human cells. Weinert and Hartwell described the mitotic spindle checkpoint as a cell-cycle checkpoint [44]; for instance, DNA damage in the G1 or G2 phase of the cell cycle generally requires cell-cycle arrest, such as the TP53 tumor-suppressor gene in mammalian cells, which then permits the repair of genotoxic damage. If the damage is permanent, it may inhibit cell proliferation or cause apoptosis [45]. Genetic changes in mitotic spindle checkpoints can directly cause chromosomal instability. If the damaged chromosome contains recessive mutations in tumor-suppressor genes, aneuploidy may significantly impact cell proliferation and the tumorigenic capacity [46]. Compared with other types of cancer, PRAD exhibited the highest genetic change frequency of *DBNDD1*, at approximately 5% (Figure 5A). This suggests that *DBNDD1* gene expression may affect mitotic spindle checkpoints and progression of PCa via the frequency of genetic alterations. In addition, molecular and physiological components and pathways connected to *DBNDD1* gene expression are intricately linked to cytoskeletal proteins, which are subfamilies of proteins such as microtubules, actin, and intermediate filaments. They are crucial for survival and cellular processes in both normal and cancer cells [47].

E2F1 plays a dual role in tumor growth, where rising E2F1 can either cause apoptosis or stimulate tumor development and invasion [48]. Overexpression of HES6 was adequate to sustain normal tumor development after castration. A ChIP-seq analysis of E2F1 in LNCaP cells revealed that HES6 enhanced the average E2F1 binding activity in the appearance of bicalutamide. HES6 may enhance E2F1 and AR activity in androgen-depleted conditions [49]. We also found that AR ligand treatment (bicalutamide) of LNCaP cells can improve E2F1 binding on the transcription start site of the *DBNDD1* gene. In addition, RB depletion increases AR binding to regions near E2F1-associated motifs, suggesting the prospect of interaction between AR and E2F1 in RB-deficient disease, and advanced stages [50]. In our finding, E2F1 binding to the *DBNDD1* transcriptional start site may be enhanced without RB in LNCaP cells. According to transcription factor-binding sites (Figure 6), *DBNDD1* expression was regulated by AR, and a number of representative transcription factors were uncovered, such as the EBF1, PAX5, BCL11A, MYC, FOXA1, FOXA2, and CEBPB, which bind and regulate *DBNDD1* gene expression. Kaiyan and colleagues found that long intergenic non-coding RNA 00844 (LINC00844) inhibited PCa cell proliferation and induced apoptosis by increasing GSTP1 via attracting EBF1 [51]. FOXA2 mutations encode a transcription factor found in neuroendocrine tumors [52], MYC-N may promote the appearance of the NEPC phenotype, FOXA1 is downregulated in NEPC, and its absence in cell line models of PCa triggers neuroendocrine differentiation. Alterations in the ratio of FOXA1:FOXA2 can induce AR cistrome reprogramming that promotes the NEPC phenotype [53]. In a test of the role of CEBPB (C/EBPβ) in PCa cell autophagy compliant with bortezomib treatment, the authors detected a reduction in the tumor growth rate of PC3 cells expressing shCEBPB, and reducing C/EBPβ enhanced sensitivity to bortezomib treatment in vivo experiments [54]. Some of the transcription factors mentioned above are known to directly or indirectly influence PCa development. They may serve as upstream regulators of *DBNDD1* gene expression. Consequently, we now have additional evidence implying a relationship between *DBNDD1* and PCa progression.

The MetaCore analysis suggested that regulating GSK-3β in bipolar disorder was closely correlated with the *DBNDD1* gene and its co-expressed genes in the progression of PCa. These results are also linked to binding terms of the GSEA results associated with GSK-3 activity, such as glycolysis and PI3K-AKT-mTOR signaling. In benign prostatic tissues, the GSK-3β protein is infrequently traceable in epithelial cells but can be detected in stromal cells. In contrast, stromal cells do not express GSK-3β, although cytoplasmic GSK-3β expression was found in approximately 30% of PCa cases with high Gleason scores (>7) [55]. GSK-3 accumulation in the cytoplasm was significantly connected with a prosurvival mechanism that fosters the formation and progression of PCa [56]. An inhibitor of GSK-3 may act as a possible agent for PCa prevention. However, the mechanisms underlying GSK-3 inhibition-mediated cancer suppression are not entirely understood and may be pretty complex, as GSK-3 is engaged in many cellular processes and has various functions [55].

In our findings, *DBNDD1* gene expression was negatively correlated with the infiltration of CD8^+^ T cells and Treg cells in PCa (Figure 9B). That means the higher *DBNDD1* expression is connected with lower infiltration of these T cell subsets. This is consistent with the trend of decreased T cell infiltration observed in PCa tissue samples [57]. Many studies were conducted over the past decade to investigate the involvement of the innate and adaptive immune systems in cancer due to the promotion of new immunotherapies for cancer. There are controversial aspects to offering immunotherapies to PCa patients. For instance, this therapy suggests the targeting of appropriate individuals; hence, these patients will respond efficiently. However, it was also indicated that determining the time of an immune intervention with this therapy and the disease stage is more important than identifying subjects. The lack of T cells in advanced tumors indicates we have been focusing on the wrong condition set. Thus, immunotherapy was suggested to be provided immediately after surgery and radiation therapy to prevent the disease from recurring. On the contrary, immunotherapy is recommended prior to hormone therapy. Thus, testosterone levels may decline, T lymphocytes may infiltrate the prostate gland, and inflammatory cytokines may be secreted [58,59,60].

Macrophages can account for more than 50% of tumor-infiltrating immune cells [61]. Typically, tumor cells instruct tumor-associated macrophages (TAMs) to evade the immune system and promote angiogenesis, tumor development, and metastasis. The polarization of macrophages (M1 and M2) is influenced by clues in the TME, such as a low pH, hypoxia, and the extracellular matrix. TAMs are thus responsible for destroying tumor cells and supporting tumor development. M2 macrophages increase tumor growth and metastasis and contribute to the poor prognoses of diseases. M1 macrophages are typically regarded as tumor-killing and immune-boosting macrophages [62]. Consistent with our findings, the *DBNDD1* gene functions as an oncogene, and its expression was positively correlated with M2 macrophages induction and negatively correlated with M1 macrophages induction. Over the past decade, immunotherapies have yielded partially favorable outcomes in various diseases, including leukemia, and kidney and skin cancers. They confront significant obstacles when applied to PCa [63]. Hence, the optimal approach for managing tumors is not to eliminate TAMs but to transform M2 TAMs into M1 antitumor macrophages [62]. Our results showed that *DBNDD1* gene expression was positively correlated with B cells and CD4^+^ effector memory T (EM-T) cell infiltration in PCa patients. We currently understand less about the link between CD4^+^ EM-T cells and PCa. We may not precisely know B cell alterations, which are a part of affinity maturation or preventing autoimmunity.

In a study of B-cell subpopulations and various immunological deficits, Saudi and colleagues observed a substantial increase in CD19^+^ B cells in peripheral blood mononuclear cells of PCa patients with lymph node metastasis relative to those without lymph node metastasis. Activated B cells were significantly induced in PCa patients’ sentinel lymph nodes that drain the tumor. The increases in switching to memory cells and plasmablasts, along with the presence of clonally enlarged B cells, demonstrate that tumor-specific T cell-dependent responses from B cells play a crucial role in B cells’ ability to combat tumors [64]. In addition, *DBNDD1* gene expression was positively connected with myeloid-derived suppressor cells (MDSCs) and negatively correlated with CD8^+^ T cell infiltration in PCa. These findings are consistent with the observation that tumor-infiltrating polymorphonuclear (PMN)-MDSCs inhibited the antitumor potential of multifunctional CD8^+^ T cells. Thus, lowering the frequency of intra-tumoral-PMN-MDSCs may inhibit PCa progression by restricting IL-23 levels, significantly contributing to PCa transformation to CRPC [65,66]. In other research, ARID1A was suggested to be a tumor suppressor in PCa, and to play a crucial role in MDSC recruitment, and the IKKβ/ARID1A/NF-kB feedback axis integrated inflammation and immunosuppression to promote PCa progression. Together, the enhanced recruitment of MDSCs has the potential to promote carcinogenesis. Inhibition of MDSCs may be a promising therapeutic option for PCa patients [67].

Based on the bioinformatics analysis, our findings are the first to determine both upstream and downstream regulatory mechanisms of *DBNDD1* gene expression in PCa. The biological processes associated with the role of *DBNDD1* gene expression in this study are depicted in a schematic diagram below (Figure 10). The constraint is that we have yet to undertake more-thorough verification by in vitro and in vivo experiments. Findings of the bioinformatic analysis can serve as a foundation for future research. DBNDD1 can potentially be a predictive biomarker for immune infiltration in PCa development.

## 4. Materials and Methods

### 4.1. Differential Gene Expression Analysis

DEGs between primary tumor tissues (498 samples) and solid normal tissues (67 samples) were processed by limma-voom [68,69]. Normalization techniques (Z normalization) were used to turn raw read counts into informative gene expression measurements and exclude factors that can influence the study. Each row in the differential gene expression table represents a gene, and each column displays the estimated differential expression measures. A volcano plot is an interactive scatter plot that illustrates the log10-fold change and statistical significance of each gene as determined by the differential gene expression analysis.

### 4.2. Enrichment Analysis

The TCGA (n = 494) was retrieved from the cBioPortal platform database (https://www.cbioportal.org accessed on 12 December 2022), which was used for data collection [70,71]. Enrichment analysis (https://maayanlab.cloud/Enrichr/ accessed on 15 February 2023) is a computerized approach for calculating details regarding an input gene set by matching it to designated gene sets representing prior biological knowledge. GO is a well-known bioinformatics endeavor designed to represent a gene’s properties across all species [72,73,74]. The figures comprise interactive bar charts depicting Enrichr-generated GO enrichment analytical results. The *x*-axis represents the −log10 (*p* value) for every term. Many relationships between these pathways and genes are documented in *KEGG*, Reactome, and WikiPathways databases [75,76,77]. Enrichr can find biological processes and pathways that have significant representation in upregulated and downregulated genes determined by assessing two sample groups. Based on the experimental results, the MetaCore (https://portal.genego.com accessed on 24 February 2023) was then utilized to determine a more profound comprehension of validated biological pathways (Omics data) [78,79,80]. 

### 4.3. Identification of Correlations between Gene Expressions and Immune Cell Infiltration

Negative and positive correlations between *DBNDD1* expression and the number of immune cell infiltrates in PCa, including those of B cells, CD4^+^ effector memory T cells, CD4^+^ type 1 helper T cells, M2 macrophages, and myeloid-derived suppressor cells, mast cells, neutrophils, CD8^+^ T cells, Treg cells, follicular helper T cells, natural killer (NK) T cells, monocytes, M1 macrophage, myeloid dendritic cells activated, natural killer (NK) cells, endothelial cells, cancer-associated fibroblasts, and hematopoietic stem cells were analyzed using PRAD databases. Adjustments were made for Spearman’s correlations and purity. Immune infiltration was evaluated by the TIMER, TIDE, EPIC, MCP-COUNTER, CIBERSORT, CIBERSORT-ABS, XCELL, and QUANTISEQ algorithms provided by TIMER 2.0 (http://timer.cistrome.org/ accessed on 10 January 2023) [81,82,83]. 

### 4.4. Survival Curve Analysis

Survival charts were obtained using the online GEPIA tool, a newly designed interactive web server to examine RNA sequencing expression data of 9736 tumor samples and 8587 normal samples from TCGA and GTEx projects [84]. There were 275 PCa samples, 182 liver cancer samples, and 239 lung cancer samples available for each studied cancer type. We also utilized it to investigate the importance of the *DBNDD1* gene in disease-free survival (DFS) in these three cancer types. The hazard ratio (HR) with a 95% confidence interval (CI) and log-rank *p* values were utilized to establish the significance of *DBNDD1* expression in these cancers.

### 4.5. UALCAN

UALCAN (http://ualcan.path.uab.edu accessed on 10 December 2022) is a friendly-to-use, interactive web resource for clinical data of approximately 30 cancer types and for assessing cancer. OMICS data built on PERL-CGI with high-quality visuals utilizing Javascript and CSS. The RNA-Seq by Expectation Maximization (RSEM) algorithm computed this database’s expression values for 20,502 genes [85]. Transcripts per million (TPM) were utilized to determine the statistical significance of variations in gene expression levels across groups. With this platform, 52, 50, and 59 normal samples and 497, 371, and 515 primary PRAD, LIHC, and LUAD samples, respectively, were obtained from TCGA database. Levels of messenger (m)-RNA for *DBNDD1* in PRAD, LIHC, and LUAD, as well as their relationships with clinicopathological features and tumor stages, were examined in this study.

### 4.6. The TNM Plot

The TNM plot web tool (https://tnmplot.com/analysis/ accessed on 20 December 2022) contains 56,938 unique samples, including Genechip from GEO (3691 normal, 29,376 tumors, and 453 metastasis), RNA-Seq from GTEx (11,215 normal), RNA-Seq from TCGA (730 normal, 9886 tumors, and 394 metastasis), and RNA-Seq from TARGET (12 normal, 1180 tumor, and one metastasis) [86]. The Normal, Tumor, and Metastatic analysis offers specifics of a chosen gene in a chosen tissue type using gene chip-based data and Dunn’s test-based statistical significance, with *p* < 0.05 (*), *p* < 0.01 (**), *p* < 0.001 (***). Statistical significance was based on Spearman correlation at *p* < 0.05 (*), *p* < 0.01 (**), *p* < 0.001 (***) for the correlation between particular genes utilizing different correlation methods in a given tissue type using gene chip data.

### 4.7. Database of Human Transcription Factor (TF) Targets

The hTFtarget comprises a comprehensive database of TF-target relationships for humans and serves as a one-stop shop for TF-target regulation research [26]. ChIP-Seq data processing and transcription factor-binding site (TFBS) scanning are used by hTFtarget to detect TF-target regulations. The hTFtarget comprises 4377 datasets from 7190 ChIP-Seq samples, 659 of which are human TFs. The ChIP-Seq data were gathered from public sources such as the NCBI GEO, SRA, and ENCODE databases. We utilized the University of California Santa Cruz (UCSC) Genome Browser, which is a user-friendly online tool for browsing genomic data and lined up annotation “tracks” in one interface [87].

### 4.8. RNA-Seq Database in Gene Expression Omnibus (GEO)

The GEO is a database repository that contains a significant amount of openly accessible gene expression data. RNA-seq and ChIP-seq of cell lines can be assessed on the GEO repository (GSE220942, GSE217260, GSE94958, GSE49832, GSE3325, and GSE55945) [49,88,89,90,91,92,93,94]. GEO2R is a dynamic web application that enables users to compare two or more groups of samples in a GEO series to figure out differentially expressed genes through experimental conditions [95], as well as the online platform available at http://www.bioinformatics.com.cn/srplot accessed on 12 December 2022. In this function, we viewed a specific gene expression profile graph by entering a gene symbol of the Human.GRCh38.p13.annot.tsv.gz annotation file [96]. 

### 4.9. Cell Culture

We obtained cell lines from ATCC, including AR-positive PCa cell lines (LNCaP and C4-2), AR-independent PCa cell lines (PC3 and LASCPC-01), and normal prostate epithelial cell line (PZ-HPV-7). LNCaP, C4-2, and PC3 cells were cultured in RPMI-1640 medium (ThermoFisher Scientific, 11875-085, New York, NY, USA) containing 5% fetal bovine serum (FBS; EMD Millipore, TMS-013-BKR, Billerica, MA, USA) and 1% penicillin. NEPC-like LASCPC-01 cells were cultured in RPMI-1640 medium supplemented with 10 nM hydrocortisone (Sigma-Aldrich, H0888, Burlington, NJ, USA), one vial insulin/transferrin/selenite (ThermoFisher Scientific, 41400-045), 200 nM-estradiol (Sigma-Aldrich, E2758, Burlington, NJ, USA), 5% FBS, and 1% penicillin. PZ-HPV-7 cells were cultured in keratinocyte serum-free medium (K-SFM; ThermoFisher, 17005-042) containing 0.05 mg/mL bovine pituitary extract (BPE; ThermoFisher) and 5 ng/mL human recombinant epidermal growth factor (EGF; ThermoFisher). Six months of AR inhibition was performed using an AR antagonist with 20 µM enzalutamide (MDV3100; Selleckchem, S1250, Houston, TX, USA).

### 4.10. Reverse-Transcription (RT)-Quantitative Polymerase Chain Reaction (qPCR)

For the isolation of total messenger (m)RNA, a RNeasy Midi Kit (Qiagen, 74004, Hilden, Germany) was used. One gram of total mRNA was prepared with the iScriptTM cDNA Synthesis Kit (Bio-Rad, 1708890, Hercules, CA, USA) for reverse transcription. For amplification, the iTaq Universal SYBR Green Supermix (Bio-Rad, 1725120) was utilized. All primer pairs were reacting on a thermocycler with an initial 10 min incubation at 95 °C, then 40 cycles at 95 °C for 15 s and 60 °C for 1 min. All reactions were conducted in triplicate and were normalized to the expression of human 18S ribosomal (r)RNA.

### 4.11. Statistical Analysis

The online GEPIA tool was used to obtain patient data and explore the effects of the *DBNDD1* gene on overall survival (OS) [84]. TCGA Pan-cancer Atlas, from cBioPortal (https://www.cbioportal.org/dataset) was accessed on 12 December 2022 [71]. Connections between DBNDD1 expression and tumor immune cells were determined using continuous default settings. A statistically significant log-rank *p* < 0.05 was chosen. 

## 5. Conclusions

In summary, DBNDD1 displays significant biological roles in PCa, and DBNDD1 overexpression can indicate a poor prognosis for PCa. Moreover, DBNDD1 can be a predictive biomarker for immune infiltration in PCa development. The findings of the bioinformatic analysis can serve as the foundation for future research.

## Figures and Tables

**Figure 1 ijms-24-11930-f001:**
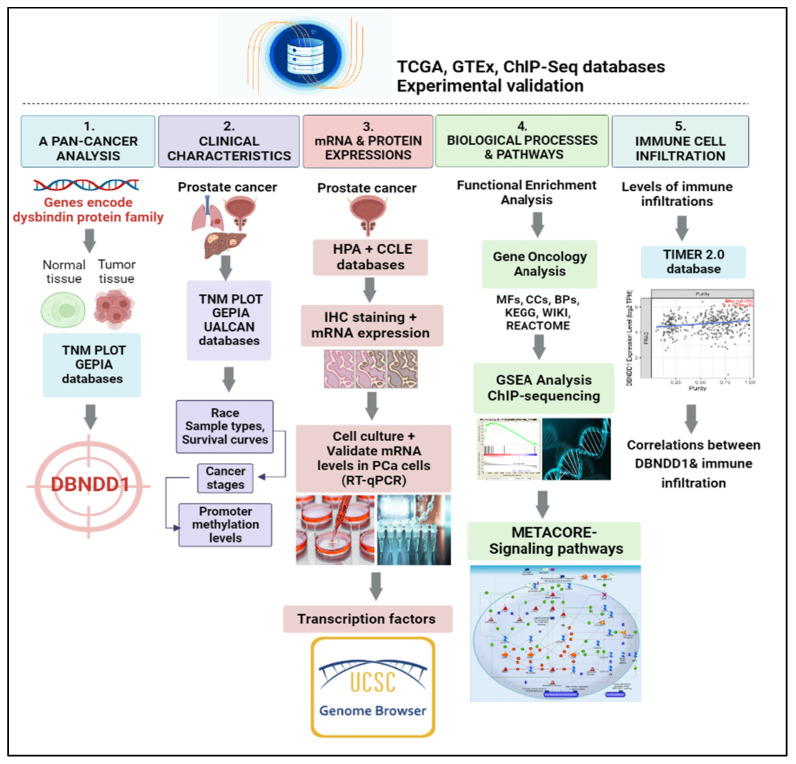
Workflow of the study design and analytical procedures. TCGA: The Cancer Genome Atlas; GTEx: The Common Fund’s Genotype-Tissue Expression; ChIP-Seq: chromatin immunoprecipitation with massively parallel DNA sequencing. TNM plot, a web tool for comparisons of Gene Expression in Normal, Tumor, and Metastatic Tissues; GEPIA: Gene Expression Profiling Interactive Analysis; DBNDD1: dysbindin domain-containing 1; GSEA: gene set enrichment analysis; UALCAN: The University of Alabama at Birmingham Cancer data analysis Portal; HPA: The Human Protein Atlas; CCLE: Cancer Cell Line Encyclopedia; IHC: immunohistochemistry; RT-qPCR: reverse transcription-quantitative polymerase chain reaction; BP: biological process; MF: molecular function; CC: cellular component; KEGG: Kyoto Encyclopedia of Genes and Genomes; TIMER: Tumor Immune Estimation Resource.

**Figure 3 ijms-24-11930-f003:**
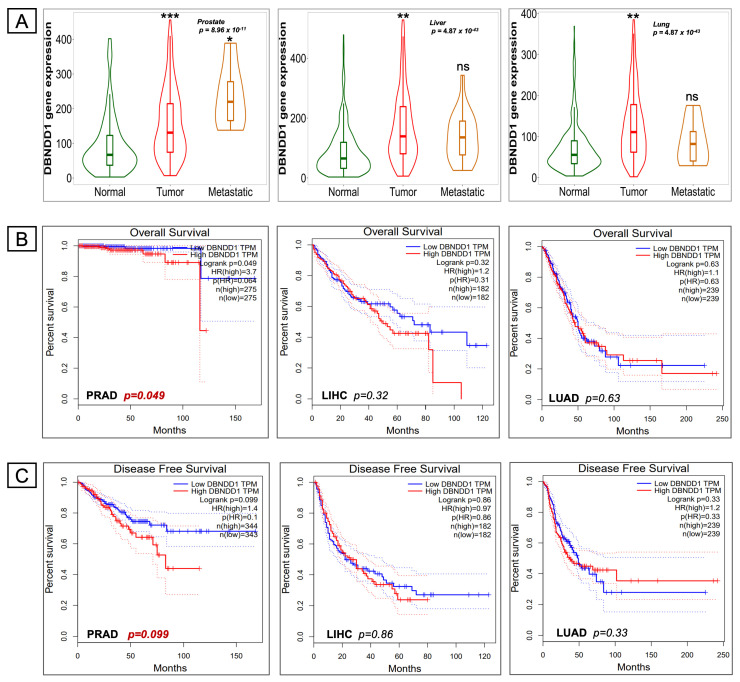
Survival analysis of *DBNDD1* gene expression in prostate, liver, and lung cancers. (**A**) *DBNDD1* expression was analyzed in Normal, Tumor, and Metastatic tissues of the prostate, liver, and lung using gene chip-based data in TNM plot. Statistical significance determined by Dunn’s test, (*) indicates a significant difference with *p* < 0.05, (**) indicates a significant difference with *p* < 0.01, (***) indicates a significant difference with *p* < 0.001, (ns) indicates non-significant. (**B**,**C**) Curves of overall survival (OS) and disease-free survival (DFS) comparing high and low DBNDD1 expressions in three types of cancer in the GEPIA webtool databases. Statistical significance determined by the Mantel-Cox test, the cox proportional hazard ratio and the 95% confidence interval.

**Figure 8 ijms-24-11930-f008:**
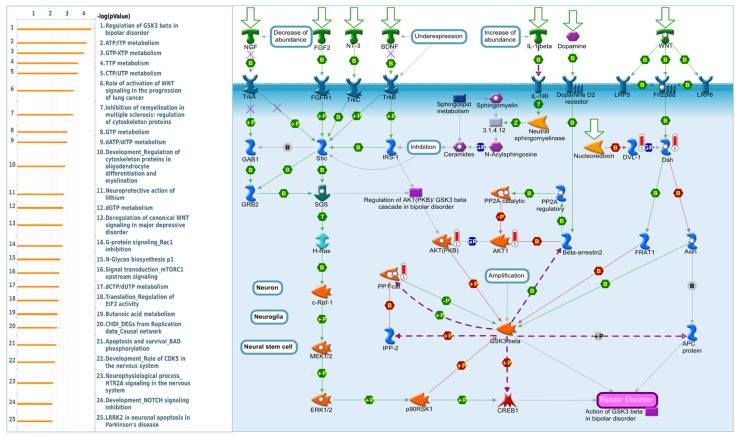
Expression of the *DBNDD1* signaling pathway in prostate cancer (MetaCore). Using the MetaCore platform to analyze genes co-expressed with *DBNDD1* from the associated TCGA dataset, we found that “Regulation of GSK3-beta in bipolar disorder” was linked to prostate cancer progression (with *p* < 0.05 set as the cutoff value).

**Figure 9 ijms-24-11930-f009:**
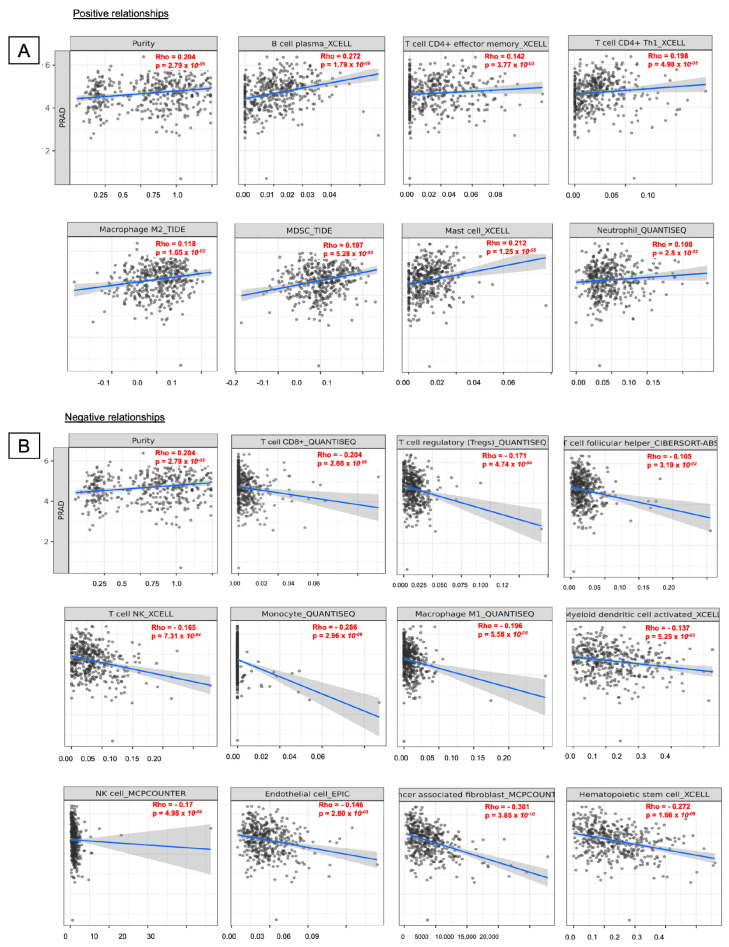
Relationships between DBNDD1 expression and immune cell infiltration in prostate cancer (TIMER). The horizontal axis represents DBNDD1 expression (values represented as log2 RNA-sequencing by Expectation Maximization (RSEM)); the vertical axis represents tumor-infiltrating immune cell markers. (**A**) Spearman’s values showed positive relationships between DBNDD1 and immune infiltration cells (B cells, CD4^+^ effector memory T cells, CD4^+^ type 1 helper T cells, M2 macrophages, and myeloid-derived suppressor cells, mast cells and neutrophils in prostate cancer) using TIMER2. (**B**) Spearman’s values showed negative relationships between DBNDD1 and immune infiltrating cells (CD8^+^ T cells, Treg cells, follicular helper T cells, natural killer (NK) T cells, monocytes, M1 macrophage, myeloid dendritic cells activated, natural killer (NK) cells, endothelial cells, cancer-associated fibroblasts and hematopoietic stem cells) in PCa, using TIMER2. The correlations between the *DBNDD1* gene and the aforementioned immune cells were described using Spearman correlations (*p* < 0.05 was considered statistically significant).

**Figure 10 ijms-24-11930-f010:**
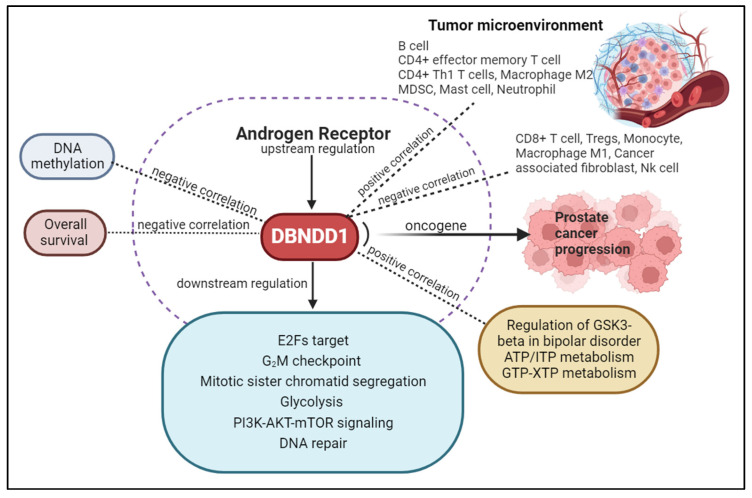
A schematic summary of the role of *DBNDD1* gene in prostate cancer progression. This image was created by BioRender.com.

## Data Availability

The datasets used and/or analyzed during this study are available upon reasonable request from the corresponding author.

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
