# Peer review of "Dysbindin Domain-Containing 1 in Prostate Cancer: New Insights into Bioinformatic Validation of Molecular and Immunological Features"

_ijms, 2023, doi:10.3390/ijms241511930_

Round 1

Reviewer 1 Report (New Reviewer)

In the present manuscript, authors have utilized different publically available datasets and analyzed the expression of DBNDD1 in different types of cancer, its impact on survival, immune infiltration, etc. The study is extensive, but it is diffused, and no experimental proofs have been provided. Alternative cancer databases should be utilized to validate their findings. The effect of DBNDD1 on cancer cells physiology/growth, migration etc needs to evaluated.

Author Response

Reviewer 2 Report (New Reviewer)

A comprehensive analysis of the role of DBNDD1 in prostate cancer. The authors examine the gene expression, p-value. FDR, Survival curve using GEPIA website, REACTOME, GO terms and KEGG patheay analysis, Relative mRNA expression with some other control genes, protein expressions, etc.. the finding was DBNDD1 is upregulating in prostate cancer. I enjoyed reading this manuscript. However, I have minor suggestions:

-Figure 4, C,D While GEPIA generated these figures, I think the approach is called Kaplan-Meier ( it is type of survival analysis method). so I suggest changing it to Kaplan-Meier curves.

-In the introduction, the authors survey the diagnostic biomarkers in the following paragraph. They may add prognostic as well. I suggest highlighting PMID: 30890858 and/or PMID: 31835700.

"Bioinformatic analyses utilizing transcriptomics and high-throughput sequencing can aid in the identification of specific cancer diagnostic biomarkers and gain a deeper understanding of molecular pathways [6]."

Author Response

Reviewer 3 Report (New Reviewer)

In this report the authors investigate the role of DBNDD1 in prostate cancers. They find some correlation between this gene and prostate cancer pathophysiologies and other expression of other genes. There are some findings here but the paper reads more like a screen than a report. There are to many figures to understand what the main point of the paper is. The title is also deceiving as it references prostate cancers but the authors report interesting findings in several cancers. Why restrict the analysis to prostate cancers? Further, there are some figures as I point below that are thoroughly supported by statistical analysis. The authors should make an effort to condense their data to 4-5 figures that present the main points and relegate the rest to supplemental. Further, the authors should produce a conclusion type figure to highlight the main findings. In its present form the paper is unnecessarily burdened with data and it is hard for a reader, including myself who specializes in querying big datasets, to understand what the main point is. Specific comments below.

Comments

1.       For figure 2 what is the gene expression value (RPKM, Z-values, etc)? The methods indicate that Z-normalization has been done but the values seem too high. Please describe in methods the exact math done to do the transformation of the TCGA values for these graphs. Additionally, what statistical tests were done to determine significance? State these things in the legend of the figure as well.

2.       In figure 3, expression levels are compared between normal tissue and various cancers in different races. However, why is normal tissue only shown once? Which normal tissue is this, Caucasian, Asian, African-American? Each race specific cancer should be compared with its corresponding normal tissue. Otherwise, the normal tissue is an amalgamation of all tissues. AS it stands the conclusion that gene expression, methylation, etc is affected by race is not supported by this analysis. Further, why are the authors using the designation “African-American” and not “black”? Are the authors absolutely sure that the patients here are black people living in the US? Generally, the term “African-American” is restricted to black people living in the US and not all black people in the world. Finally, the statistical test used to determine significance should be indicated in the legend.

3.       For figure 5C a better representation should be drawn. What domains of dysbindin are affected by that truncation? Is the truncation heterozygous or homozygous? Also, the text says that mutations are depicted, but the authors show only one mutation. Is this a hotspot mutation, or is there only one mutation? Why not depict the other ones?

4.       Section 2.5 and accompanying figure. The authors investigate protein levels but call it over-expression. Then the authors discuss gene expression levels which I assume this time it refers to mRNA (e.g. transcription). The authors conclude that “it might be higher in LNCaP…” This is confusing. Are the authors suggesting that an increase in transcription correlates with an increase in protein levels? Or is translation independent of transcription (e.g. an increase in protein levels does not correlated with an increase in mRNA levels). If the whole point of this section is to differentiate transcription from translation, this fact does not come through in this section. Additionally, at line 260 the authors say that “mRNA expression might be considerably higher…”. This is not scientifically sound: if it is “might” this indicates that they are not sure, if it is “considerably” then it is a sure fact. Only statistics can show this. Please run statistics to determine if the correlation exists or not.

English is fine

Round 2

Reviewer 1 Report (New Reviewer)

In the present manuscript, authors have explored the association between Dysbindin Domain-Containing 1 expression and different molecular and immunological features of different types of cancers, including prostate cancer tumors, by utilizing different publicly available databases. Further, they evaluated the expression of this gene in different prostate cancer cell lines. Authors have demonstrated that DBNDD1 expression positively correlates with prostate cancer tissue immune infiltration and may serve as early-stage prognostic markers. The authors have performed extensive analysis and correlated the DBNDD1 expression with different oncogenic features. The manuscript has significantly improved after revision. I have the following concerns:

1- Potential molecular regulators of DBNDD1 need to be discussed in the discussion section.

2-Font size of labels in figures needs to be increased.

3-Name of the statistical test needs to be mentioned in figure legends.

4- The Introduction section needs to be more precise, focusing on prostate cancer.

5-In Figure 4I, it will be good to present IHC intensity as percentage.

Author Response

Reviewer 3 Report (New Reviewer)

The authors have made the changes I requested and clarified some of the language in the text. Most importantly, I appreciate the introduction of a conclusion figure to tie the data together.

Author Response

This manuscript is a resubmission of an earlier submission. The following is a list of the peer review reports and author responses from that submission.

Round 1

Reviewer 1 Report

Reviewer’s Comments

The manuscript “New Insight into the Immunological and Prognostic Significance of Genes Encoding the Dysbindin Protein Family in Prostate Cancer” is a very interesting work. In this work, Prostate cancer (PCa) is one of the most prevalent cancers in men, yet its pathogenic pathways remain poorly understood. Transcriptomics and high-throughput sequencing could help us uncover cancer diagnostic targets and understand biological circuits. Using the prostate adenocarcinoma datasets of the Gene Expression Profiling Interactive Analysis (GEPIA) and web-based applications (UALCAN, cBioPortal, Xenabrowser, SR Plot, hTFtarget, Genome Browser, MetaCore), we found that up-regulated DBNDD1 expression in primary prostate tumors was strongly correlated with issues involving cell cycle, mitotic homo sapiens in KEGG, WIKI, and REACTOME pathways, and transcription factor binding sites with DBNDD1 gene in prostate samples. The DBNDD1 gene expression was significantly influenced by race, sample type, cancer stage, and promoter methylation levels of different cancers such as prostate adenocarcinoma (PRAD), liver hepatocellular carcinoma (LIHC), and lung adenocarcinoma (LUAD). The MetaCore analysis revealed that regulation of GSK-3β in bipolar disorder and metabolic pathways correlated closely with the DBNDD1 gene and its co-expressed genes in the PCa progression.  While I believe this topic is of great interest to our readers, I think it needs major revision before it is ready for publication. So, I recommend this manuscript for publication with major revisions.

1. In this manuscript, the authors did not explain the importance of Immunological in the introduction part. The authors should explain the importance of Immunological.

2) Title: The title of the manuscript is not impressive. It should be modified or rewritten it.

3) Correct the following statement “DBNDD1 gene expression was increased in various kinds of cancer, affecting prognosis and immune infiltration in PCa patients. We are the first to use bioinformatics and data mining to investigate the transcription levels and biological functions of the DBNDD1 gene in PCa”.

4) Keywords: The Immunological is missing in the keywords. So, modify the keywords.

5) Introduction part is not impressive. The references cited are very old. So, Improve it with some latest literature such as 10.3390/molecules27217368, 10.3390/molecules27207129

6) The authors should explain the following statement with recent references, “To investigate the clinical characteristics associated with DBNDD1 expression, we examined whether race, sample type, cancer stage, and promoter methylation level influence DBNDD1 expression in three of the abovementioned cancers”.

7) Add space between magnitude and unit. For example, in synthesis “21.96g” should be 21.96 g. Make the corrections throughout the manuscript regarding values and units.

8) The author should provide reason about this statement “To understand how the DBNDD1 gene expresses in some different types of tissues and whether its expression influences survival rate, we investigated their variations based on datasets of the TNM plot and GEPIA based-tool”.

9. Comparison of the present results with other similar findings in the literature should be discussed in more detail. This is necessary in order to place this work together with other work in the field and to give more credibility to the present results.

10) Conclusion part is very long. Make it brief and improve by adding the results of your studies.

11) There are many grammatic mistakes. Improve the English grammar of the manuscript.

Minor editing of English language required

Reviewer 2 Report

The research article being reviewed presents an in-depth analysis of the DBNDD1 gene and its potential as a biomarker for prostate cancer (PCa). The study found that DBNDD1 expression is significantly increased in numerous tumor types, including PCa and is associated with poor overall survival in PCa patients. The authors also examined the clinical characteristics associated with DBNDD1 expression, such as race, sample type, cancer stage, and promoter methylation level, in various cancers and found that these factors significantly affect DBNDD1 expression in PCa. The study further investigated the protein and transcription expressions of DBNDD1 in prostate tissues and cell lines and found that the overexpression of the DBNDD1 protein is detected in prostate tumor tissue samples and several PCa cell lines. The authors also identified several transcription factors that regulate DBNDD1 gene expression, providing additional evidence for the relationship between DBNDD1 and the progression of prostate cancer.
Overall, this research article proposed  DBNDD1 gene as a potential biomarker for PCa.  The article is well-written and utilises a variety of bioinformatics tools and datasets to provide valuable insights. However, there are concerns regarding the rationale for choosing DBNDD1 as a candidate biomarker.
The authors should clarify in the introduction and results section the basis for selecting DBNDD1 as a candidate biomarker, particularly given the lack of prior knowledge on its association with PCa. If a candidate-based approach was used, this should be clearly stated in the results section. Additionally, the authors should address how they ensured DBNDD1 is not a random target, as it is important to distinguish it from the approximately 25,000 other genes in the human genome.
Until these concerns are adequately addressed, it is difficult to have confidence in the proposed use of DBNDD1 as a biomarker for PCa. Therefore, I do not recommend publication at this time.

A minor comment regarding the manuscript is that Figure 10 table is difficult to read and appears illegible. One suggestion would be for the authors to present this information in a supplementary table instead, with the major pathways and corresponding p-values clearly labeled for easy reference.

Round 2

Reviewer 2 Report

After thoroughly reviewing the author's response, I find myself unconvinced that DBNDD1 can effectively serve as a prognostic marker for PCa. The rationale provided does not sufficiently address the concerns I raised and fails to present compelling evidence to support its selection as a reliable biomarker.
Given the critical nature of biomarker research in guiding clinical decisions and patient care, it is of utmost importance to ensure that the proposed biomarkers have a strong foundation of evidence. In the case of DBNDD1, I believe further research is necessary to establish its prognostic value and reliability.

Therefore, I cannot recommend publishing this article.